# Applying Protein–Protein Interactions and Complex Networks to Identify Novel Genes in Retinitis Pigmentosa Pathogenesis

**DOI:** 10.3390/ijms23073962

**Published:** 2022-04-02

**Authors:** Su-Bin Yoon, Yu-Chien (Calvin) Ma, Akaash Venkat, Chun-Yu (Audi) Liu, Jie J. Zheng

**Affiliations:** Department of Ophthalmology, Stein Eye Institute, David Geffen School of Medicine at UCLA, Los Angeles, CA 90095, USA; yoonsuvin@ucla.edu (S.-B.Y.); calvinma888@ucla.edu (Y.-C.M.); akaashvenkat@gmail.com (A.V.); chuliu17@g.ucla.edu (C.-Y.L.)

**Keywords:** Retinitis Pigmentosa, retinal degeneration, protein interaction network (PIN), protein–protein interaction (PPI), complex networks, network medicine, bioinformatics, STRING

## Abstract

Retinitis Pigmentosa (RP) is a hereditary retinal disorder that causes the atrophy of photoreceptor rod cells. Since individual defective genes converge on the same disease, we hypothesized that all causal genes of RP belong in a complex network. To explore this hypothesis, we conducted a gene connection analysis using 161 genes attributed to RP, compiled from the Retinal Information Network, RetNet. We then examined the protein interaction network (PIN) of these genes. In line with our hypothesis, using STRING, we directly connected 149 genes out of the recognized 159 genes. To uncover the association between the PIN and the ten unrecalled genes, we developed an algorithm to pinpoint the best candidate genes to connect the uncalled genes to the PIN and identified ten such genes. We propose that mutations within these ten genes may also cause RP; this notion is supported by analyzing and categorizing the known causal genes based on cellular locations and related functions. The successful establishment of the PIN among all documented genes and the discovery of novel genes for RP strongly suggest an interconnectedness that causes the disease on the molecular level. In addition, our computational gene search protocol can help identify the genes and loci responsible for genetic diseases, not limited to RP.

## 1. Introduction

Retinitis Pigmentosa (RP) is one of the monogenic human retinal dystrophies [1]. It is an inherited retinal disorder mainly characterized by the progressive degeneration of photoreceptor rod cells [2,3,4]. It develops over the course of several decades, but the majority of the progression occurs within the first four decades from birth [2,3,4]. Since RP is a monogenic disorder, a defect in any single one of the many causal genes produces the same outcome we collectively diagnose as RP [5]. We hypothesize that all the RP causal genes should interconnect at the molecular level to replicate the same set of symptoms.

Despite not having a complete picture of the relationship among all RP causal genes, many studies linked the genetic mutations in RP to potential mechanisms of rod photoreceptor cell death in RP [6]. For instance, ABCA4, a known RP gene, is involved in response to oxidative stress by removing toxic compounds from oxidative stress and promoting cell survival [7,8,9]. Therefore, a defect in the ABCA4 protein can cause an accumulation of oxidative stress, which in turn can trigger the inflammatory cascade leading to photoreceptor degeneration. Other mechanisms of photoreceptor cell death involve endoplasmic reticulum stress and Ca^2+^ accumulation [6,10,11,12]. Nevertheless, we are missing the puzzle pieces that would help us understand the RP mechanism in a network rather than isolated pathways that only converge at the disease outcome.

Previous computational studies have found that pathospecific genes tend to be associated with each other, creating a neighborhood of genes we call a disease module [13,14]. These modules serve as a map to understanding the pathogenesis of a disease based on molecular interactions. Currently, many human genome-wide interaction network databases are available. These complex networks demonstrate the interaction among RP causal genes in the form of nodes and links—nodes being the genes and links being the interaction [15,16]. Therefore, we decided to develop and test whether we can use a bioinformatics database search algorithm to connect all the RP causal genes based on the known protein–protein interactions (PPIs).

Among multiple databases of protein interaction networks (PIN), we decided to use STRING (Search Tool for the Retrieval of Interacting Genes/Proteins), as it gathers, assesses, and incorporates various PPI information such as gene co-expression, literature, and experimental protein–protein association data [17]. STRING’s built-in enrichment analysis uses a combination of traditional classification systems such as KEGG (Kyoto Encyclopedia of Genes and Genomes) and new methods such as high-throughput text-mining and hierarchical clustering of the association network itself. Using the data amassed, users can input a list of gene products to the STRING database to visualize the physical and functional interactions, both annotated and scored [17,18,19]. Although STRING uses the established term PPI to describe the fundamental focus of the database, this comprehensive database also integrates indirect, functional interactions of the genes [17]. Compared to other network bioinformatics tools, STRING is documented to be among the most reliable and sensitive databases [15,19,20].

Considering that the product of computational analysis is only as reliable as the known causal genes used to find the novel genes, we selected 161 genes that are verified to cause RP through the Retinal Information Network, also known as RetNet (https://sph.uth.edu/RetNet/ accessed on 15 February 2019) [21]. However, genetic disorders, including RP, are mainly studied and diagnosed by sequencing and screening the genome of an affected individual [22]. This method heavily relies upon individual reported cases to formulate a hypothesis and conduct specific research—as a response to the occurrence rather than taking a more active initiative. For a rare inherited disorder such as RP, the current dependency on clinical data for gene discovery is expensive and time consuming.

Reflecting the status quo, the known genes to date are estimated to account for less than 50% of all RP patients [3]. Since the emerging treatments of RP such as adeno-associated virus (AAV) vector-mediated gene therapy require a known mutation in a causal gene, the rest of the patients cannot benefit from the ongoing clinical trials and future treatments. Furthermore, it makes the construction of an RP disease module significantly more challenging due to the lack of available causal genes as a rare disease.

To tackle this issue, we define *intermediate genes* as candidate or novel genes that may cause RP by interacting with the already known genes causative of the disease. Each known causal gene of RP is assessed on STRING to create a dictionary of interactive genes that meet the interaction threshold. Considering that a disease is a manifestation of a specific set of symptoms, the genes that give rise to these symptoms are highly likely to interact with each other. Therefore, all causal genes of RP would be interconnected. Under this notion, the genes that lack PPI to connect with any other causal genes found on the initial list may have an intermediate gene that facilitates the interaction [18]. Hence, these genes are intermediate or novel genes that could be the missing link. Therefore, we investigate the novel genes in the pathogenesis of RP via computational analysis of PPI and evaluate them with pre-existing literature.

The causal RP genes may be studied in different contexts, such as another ocular disorder, but never be linked to RP. Therefore, these genes would never be recognized in the official database and create a knowledge gap [3]. Proceeding from an apparent overlap among causal genes for various retinal diseases, these omitted genes can be detected by scrutinizing PPIs [5]. Based on the hypothesized interconnectedness among genes, we aimed to locate those that connect the gap among the genes known to cause RP with an algorithm using PPIs. The algorithm would accelerate the genetic research by proposing candidate genes that could cause RP. At the same time, it would support the hypothesis that these genes are all connected, possibly via PPI.

## 2. Results

### 2.1. Initial Retinitis Pigmentosa Gene List and Gene Mapping

The original 161 genes collected from RetNet (https://sph.uth.edu/RetNet/ accessed on 15 February 2019) (Appendix B) were all found in unique individual patient cases, where each patient had only one dysfunctional gene [21]. The defect in those unique, individual genes eventually caused each patient to be afflicted with some form of RP. Considering the wide range of disease outcomes for RP, all genes were positively associated with the disease regardless of minor discrepancies in the severity or type of disease manifestation.

Among the 161 genes, two genes, TTC8a and UTY, were not recognized by the STRING v11 database, and they were excluded from the study. Therefore, the final 159 registered genes were used in the study. Out of these 159 genes, 2 genes were referred to differently on the STRING v11 database: SC5DL as SC5D and C5orf4 as FAXDC2.

Of the network of 149 genes constructed by STRING, 10 genes remained completely disconnected (Figure 1a). Nevertheless, the instant complex network formation of 149 genes implies that these seemingly unrelated, isolated RP causal genes are in a common network determining RP’s onset.

### 2.2. Discovery of Intermediate Genes

We then asked if we could find genes, referred to as candidate genes, which could connect the 10 uncalled genes to the 149-gene network. An algorithm to evaluate those genes based on the enrichment analysis on STRING was developed. Using the algorithm, we selected the ten best candidate genes to be the intermediate genes that complete the global network of RP genes by bridging the ten uncalled genes to the network. The ten intermediate genes to interconnect all RP-causing genes are: CDH2, EVA1A, PNPT1, PLK1, RHOA, GNG2, GNGT1, GART, ITGB2, and DOLK [5,21]. 

These intermediates, visualized as green nodes, were then mapped with the rest of the genes compiled from RetNet (Figure 1b). The white, yellow, and green nodes represent the original 159 genes and the 10 intermediate genes. The white nodes are the genes from RetNet connected directly without further processing (Figure 1b). The yellow nodes are the genes from RetNet that required an intermediate gene to connect to the greater network, previously shown to be disconnected in the absence of intermediate genes (Figure 1a,b). The green nodes represent the intermediate genes that connect their respective yellow nodes to the rest, namely the white nodes (Figure 1b). The intermediate genes discovered based on PPIs complete the connections among the documented causal genes of RP. We call these intermediate genes “candidate genes” because it is likely that they also contribute to RP.

### 2.3. Functional Analysis of Intermediate Genes with Gene Ontology

Gene ontology is a bioinformatics system that describes gene and gene product functions. Gene ontology enrichment analysis gives a general overview of biological processes, molecular functions, and cellular compartments for a given set of genes. Using the analysis provided by STRING, three intermediate genes—CDH2, GART, and RHOA—were found to be involved in cerebral cortex development (Appendix A). Of the three genes, CDH2 and RHOA are involved in radial glial cell differentiation (Appendix A). GTPase activity was the only enriched molecular function observed from the set of intermediate genes, with GNG2, GNGT1, and RHOA producing matching proteins for the function (Appendix A).

### 2.4. Gene Classification and Categorization

Based on the literature, we classified all the RP genes into four groups according to their cellular/subcellular localization. The genes were computationally categorized into these groups, thus placing them near other genes with high probabilities of forming PIN (Figure 2). The grouping more accurately represents the functional locations of the gene products. 

Figure 3 demonstrates the relationships among the groups of RP genes based on the localization of the products. Group 1 (topmost) represents genes that are involved in the retinal pigment epithelium (RPE) cells (Figure 3a), whose function is retinal metabolism [23]. Group 2 includes the gene products that take part in phototransduction in the outer segment (OS), converting light into other signals (Figure 3b). The Group 3 genes and their products partake in roles related to ciliary structure and gateway functions in the connecting cilium (Figure 3c). Group 4 represents the genes involved in transcription and splicing based on their localization to the nucleus (Figure 3d) [23]. The intermediate genes were placed with the respective disconnected gene. Depending on the origin and the target node, a different colored edge was used to show the PPI and explore inter- and intragroup interactions on a network level.

Dias et al. identified 30 genes out of the 159 RP genes to belong to one of the four aforementioned groups [23]. Based on those 30 genes, we created an algorithm to sort the rest of the genes by assigning each gene to a group that has the most direct PPIs with the said gene. When there is a tie between two groups regarding the number of connections, the gene is sorted into the group with a higher sum of overall PPI confidence levels. With each algorithm iteration (i.e., each time a gene is assigned to a group), the groups expand with a new gene sorted into them. The expanded groups allow the previously unassigned genes due to the lack of PPI with Groups 1–4 to be revisited and sorted. There are no genes left unassigned, since all genes are connected as hypothesized. 

The edges between any two given genes represent PPIs. They are annotated in colors to show their connection with respect to the groups as previously defined. The colors visualize the number of PPIs associated with each group. The red, blue, green, and purple edges connect to a gene in Group 1, 2, 3, and 4, respectively. The gray edges are used between a specific gene that connects to an intermediate gene. Further explanation can be found in Appendix C.

The PPIs among groups were counted to measure the interaction between any two groups of genes (Table 1). Although there is no suggestible trend, it is noteworthy that Group 2 contained the greatest number of connections with the intermediate genes discovered by the protocol. Group 2 is attributed to a sub-localization of the cell, OS, which converts the light signals into vision as we understand it by translating the rhodopsin photoisomerization into electric signals. Group 4 gene products involved in transcription and splicing localize in the nucleus. Group 4 has interactions with all other groups, indicating the possibility of modulated gene expression within the network that results in the RP phenotype.

## 3. Discussion

### 3.1. Summary

Our study demonstrates an interconnectedness of the genes that cause RP through the successful establishment of PPIs among all documented genes. The initial network formation by 149 genes, as well as the discovery of novel genes, strongly support our hypothesis of the global connectedness of the genes that cause the disease. As with disease modules, the complete connection suggests a possible pathway that converges on a molecular level. The greatest number of PPIs was found in Group 2 (OS), the group of genes responsible for transforming rhodopsin photoisomerization into electric signals for the brain to interpret. These PPIs point to a potential convergence point within the pathway that may be a part of the RP mechanism leading to blindness. Indeed, the photopigment rhodopsin, encoded by RHO, is a prerequisite for photoreceptor cell viability and vision [24]. Furthermore, the previous literature (refer to Section 3.2) on the association of novel genes to RP supports the hypothesis that all genes that cause RP must be interconnected. 

### 3.2. Overview of Candidate Genes’ Potential Roles in Retinitis Pigmentosa Pathology

RHOA, one of the ten candidate genes, is a member of the Rho family that encodes GTPase proteins heavily involved in cellular signaling. Gene ontology enrichment has indicated that RHOA is a part of the GTPase activity process along with two other genes, GNG2 and GNGT1 (Appendix A). As a crucial regulatory gene, RHOA appears in multiple studies of retinal degeneration or dystrophy in relation to other genes. RPGR is one of the genes found to cause RP when mutated [25]. The transcriptomics data of a rod-dominant mouse retina with an RPGR knockout showed differential regulation of genes that encode for regular actin cytoskeletal dynamics, as well as an increased expression of RHOA-GTP, indicating a correlation between them [25]. Genes related to RHOA, STARD13, and RTKN2 were also overexpressed [25]. STARD13 regulates RHOA, and RTKN2 binds to the activated form of RHOA as an effector, hence supporting that an increased expression of RHOA is a part of the RP pathogenesis in conjunction with RPGR [25,26,27]. 

Drawing from RHOA’s role as a GTPase protein encoder, the two other genes found to be involved in the GTPase activity by gene ontology enrichment analysis are also likely to have a similar effect on the pathogenesis of RP (Appendix A). As it turns out, GNG2 and GNGT1 are both G protein gamma subunits that are significant in signal transduction (Appendix A) [28]. In particular, GNGT1 is a gamma subunit of transducin, a guanine nucleotide-binding protein (G protein) which localizes in rod outer segment [29,30,31]. Not only does our computational categorization of GNGT1 into Group 2, the OS, match the literature, it also suggests a role in RP (Figure 2 and Figure 3b). A type of transducin, also known as GMPase, interacts with rhodopsin to activate a cyclic GTP-specific phosphodiesterase. In relation to RHOA and GNGT1, GNG2 is also likely a novel gene that can cause RP [29].

GART, another candidate gene, encodes an enzyme that participates in multiple steps of the inosine monophosphate (IMP) synthesis pathway. Because IMP synthesis is upstream of the ATP and GTP synthesis pathways, a defect in the upstream pathway such as GART can affect ocular development [32]. The authors have found that perturbation in the retinoblast development due to such a defect results in microphthalmia [32]. Considering the literature on purine-mediated signaling and ocular development, it is plausible that a defect in GART could be a part of the RP pathogenesis [33].

GART was identified in gene ontology to be a part of the cerebral cortex development process along with RHOA and CDH2 (Appendix A). RHOA and CDH2 are also a part of the radial glial cell differentiation processes (Appendix A). One of the most prominent types of glial cells in the retina that persists through development and into the adult retina is the Müller glia [34]. Müller glial cells are involved in protecting retinal neurons, homeostasis of the retinal extracellular environment, and optical transfer, among others [34,35,36]. They are also retinal stem cells and progenitors that respond to retinal injury, and Müller glial cells have been studied to explore their regenerative function in models with retinal degeneration such as RP [37,38,39,40]. In this respect, GART, RHOA, and CDH2 all have the potential to give rise to RP.

DOLK is known to be pathogenic for a form of α-dystroglycanopathies. POMGNT1, from the initial list of 161 genes, is also known to cause the same disorder [41]. Mutations in POMGNT1 can also cause RP-76. Although concrete evidence is yet to be established for DOLK and how its defect can cause RP, defects in glycosylation are reported to cause a wide array of diseases, including RP. Several genes that participate in glycosylation, and defects in these genes such as POMGNT1 and DHDDS, have been found to cause different types of RP [42,43,44,45]. In light of this connection, DOLK has the potential to be one of such genes that cause RP through a defect in glycosylation.

### 3.3. Limitations

The outcome of a computational study such as ours depends on the input parameters. The definition of causal genes can vary, and the results may not stay consistent. It is also sensitive to new information because both the input and bioinformatic tools are subjected to updates based on the new data. However, the paper addresses this by providing the date of retrieval for the input data and the version of the tools being used to conduct the search. This allows anyone to replicate the results using the information given in the manuscript and associated supplementary tools. 

The literature search is highly subjective and should be considered an exploratory review of the genes. Despite the uncertainty in the contribution of intermediate genes in RP, the protocol is a way to draw a set of genes for further experimental research that genetically tests whether these genes are responsible for a form of RP or other genetic diseases. This not only gives an idea for genetic and clinical research via probable candidate genes, but also promotes the efficiency of genetic research to advance in a uniform direction with less trial and error. Hence, the protocol serves its purpose as a computational candidate gene identification tool.

## 4. Materials and Methods

### 4.1. Retinitis Pigmentosa Gene Compilation through RetNet

To construct an interconnected map of genes related to RP, a comprehensive list of genes known to cause the disorder was compiled using RetNet (https://sph.uth.edu/RetNet/ accessed on 15 February 2019). From this database, the name, location, and associated diseases with the malfunction of the genes were recorded on 15 February 2019. The 161 genes related to RP were compiled and can be found in Appendix B.

### 4.2. Protein–Protein Interactions and Functional Analysis of Novel Genes

For each gene from the original 161 genes (of which only 159 were recognized by STRING), the Python program uses Selenium Webdriver to automate the process of entering gene names on STRING to determine PPIs of the gene. We scraped the HTML file that results from the automation to find the gene’s PPIs and the confidence level for each connection. 

The novel genes discovered by the Python program were then analyzed based on the PPI, gene ontology information, and previous literature. The PPIs were again gathered by STRING to be reviewed in further detail. 

Gene ontology (GO), which is computable information about the function of genes and their protein products, can be extremely helpful in searching for a gene’s connection to other genes in addition to learning more about the function of a gene. The GO database used for this paper is the default database integrated into STRING, where gene ontology information is collected from countless sources and put together for a unified database for gene functions.

The primary search engine for the biomedical literature used in this paper was PubMed.

### 4.3. Python Script for Novel Gene Discovery, Classification, and Categorization

After storing the PPIs and respective confidence levels for each input gene in the Python dictionary, we used that data to determine if the input genes are directly connected or require an intermediate gene (that can be found in the dictionary) that can connect the input genes to the rest of the genes. As a result, these genes were sorted into four groups: A, B, C, D.

Group A genes are those from the input list that connect directly to other genes from the input list. If most input RP genes fall into this category, it supports the hypothesis that all RP genes are indeed interconnected. Group B genes are those from the input list that require an intermediate gene to connect to the rest of the genes from the input list. Our intermediate genes are for connecting Group B genes to Group A genes. Group C genes are those from the input list that do not connect to the rest of the genes from the input list, even with potential intermediate genes. These indicate that no global network converges on at least one level. Group D genes are the intermediate genes that connect the Group B genes with the rest of the genes from the input list. 

Essentially, group D genes interact with the genes present in the map/dictionary but are absent from the existing database for RP. We see group D genes as the missing links or the novel genes that contribute to RP but have not been identified in the original list. These genes serve as a bridge between the disconnected genes and the rest to complete one network of PPIs. Although we cannot identify the distinct pathways with confidence, we focus on the interconnectedness that these intermediate genes can bring. Note that to minimize the number of genes in the map, we use the confidence levels stored in the Python dictionary to pair only one Group D gene with each Group B gene.

After classifying the genes into Group A, B, C, or D, the program then uses Selenium Webdriver again to automate the process of entering all the genes (the original 159 genes and the additional ten intermediate genes) into STRING and to download the resulting map, in SVG format. The four groups are visualized by color on the SVG, with each group of genes having a distinct color. The map is then further modified according to cellular/subcellular localizations. There are four groups based on their function and putative localization: (1) RPE, (2) OS, (3) connecting cilium, and (4) nucleus. Group 1 genes play a role in retinal metabolism [23]. Group 2, Group 3, and Group 4 represent the genes involved in photoreceptor cells, which are crucial for phototransduction, ciliary structure, and transcription and splicing [23].

The localization of gene products is significant in determining the likelihood of PPIs among genes. In vivo, proteins that interact with each other are highly likely to reside in close proximity, usually within the same or adjacent cellular/subcellular compartments. Therefore, the function and putative localization of 30 genes were taken from previous research to extrapolate the localization of other genes: the more PPIs, the higher the likelihood of being in the same cellular or subcellular compartment. Each gene is sorted into one of the groups—established by the localizations of 30 genes mentioned above—based on the number of PPIs it has with each group. The gene is categorized into the group with the greatest number of PPIs. When there is a tie in the number of PPIs, we use the sum of PPI confidence levels as the tiebreaker. There are more genes to extrapolate from after each iteration of this algorithm. Therefore, we run this algorithm for multiple iterations until all genes are sorted by cellular/subcellular localizations.

## 5. Conclusions

In-depth knowledge of PPIs and their intricate network is the key to the understanding of interactions that are the foundation of cellular processes behind disease mechanisms. Of the ten intermediate genes discovered through PPI analysis, six genes show a possible link to RP, supported by previous research. There is no apparent relationship to RP that can be found in the other four genes. However, this is not to conclude that these genes are not a part of the disease mechanism. The discovery of intermediate genes exclusively through computational methods indicates some reliability in retrieving novel genes with a protocol developed in the manuscript. The findings suggest that significantly more genes could contribute to RP, as expected by previous literature [3]. In addition to the novel genes, the successful interconnection of the documented genes responsible for RP via PPIs supports our hypothesis that all genes that cause RP are connected to produce the same outcome.

One aspect of a future study would be to explore the mechanism for the development of RP. This manuscript explores whether the genes are interconnected based on PPI and concludes that evidence favors such a hypothesis based on our computational analysis. Therefore, future studies should focus on how these genes are connected and on what level they converge and diverge in order to give rise to the same disease despite the mutations being in entirely different genes. Another aspect of a future study would be the expansion of genetic databases with a similar computational protocol accompanied by a rigorous literature search and data mining. They can also incorporate more bioinformatic tools to increase the accuracy and precision of the protocol outlined in the manuscript. Concurrent genetic studies can then supplement this process to confirm the role of each novel gene in RP. The combination of these two research directions would undeniably solidify our understanding of this rare retinal disorder and advance the prevention and treatment of patients with this disease.

## Figures and Tables

**Figure 1 ijms-23-03962-f001:**
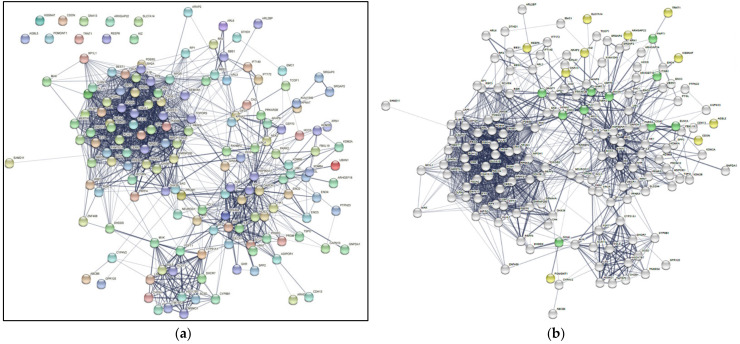
Complex PPI network of RP causal genes. (**a**) The Initial Map of Genes Based on the Protein–Protein Interactions from the STRING v11 Database. The disconnected or isolated genes without any PPI are shown in the top left corner: POMGNT1, REEP6, HGSNAT, CDON, GNA13, ARHGAP22, SLC7A14, AGBL5, TRNT1, KIZ. (**b**) The Color-coded Map of Genes Based on the Protein–Protein Interactions from the STRING v11 Database, including Intermediate genes. This map is the original output downloaded from STRING before the gene classification and automation. The white nodes represent the genes from RetNet that can connect to each other directly. The yellow nodes represent the genes from RetNet that required an intermediate gene to connect to the rest of the map, previously shown to be disconnected in the absence of intermediate genes in Figure 1a. The green nodes are the intermediate genes that were discovered by the protocol.

**Figure 2 ijms-23-03962-f002:**
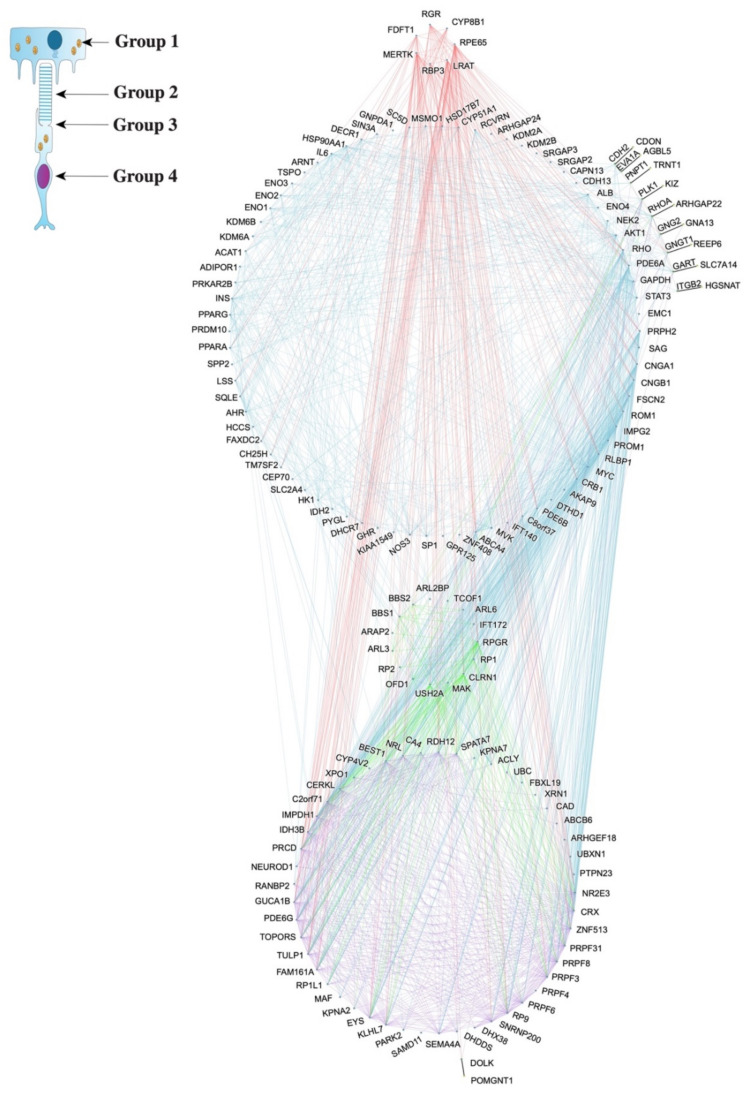
The Reorganized Complex Network of Genes Based on the Protein–Protein Interactions from the STRING v11 Database. This map has been through the gene classification and automation described in Appendix D. The genes are organized into four groups according to gene product localization and are annotated to show their connectivity. Group 1—retinal pigment epithelium (RPE); Group 2—OS; Group 3—connecting cilium; Group 4—nucleus. The intermediate genes are next to their respective disconnected gene. The color of an edge is determined based on the originating node for each interaction: red for Group 1, blue for Group 2, green for Group 3, and purple for Group 4.

**Figure 3 ijms-23-03962-f003:**
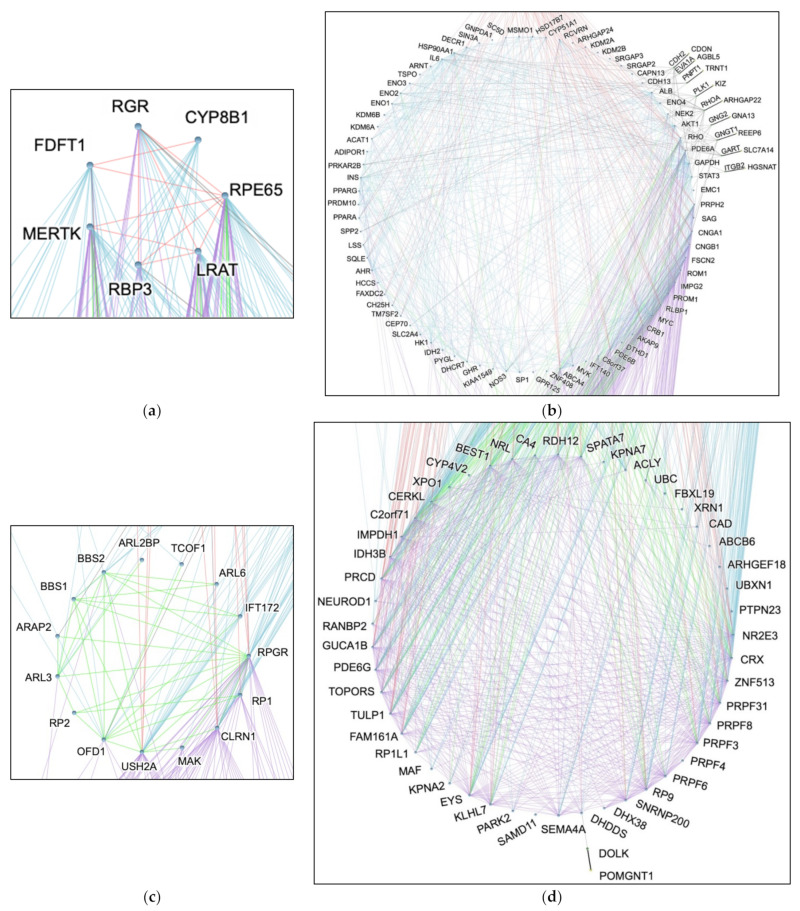
The Annotated Individual View of Complex Network of Genes Based on the Protein–Protein Interactions from the STRING v11 Database. Each group is zoomed in for a closer inspection of the relationship portrayed in Figure 2. Since PPIs are bidirectional, each edge in Figure 2 can be annotated in two different colors. In this panel, each figure is annotated with each respective group being the origin node (from: origin node, connects to: Group 1—red; Group 2—blue; Group 3—green; Group 4—purple). (**a**) Expanded view of Group 1; (**b**) expanded view of Group 2; (**c**) expanded view of Group 3; (**d**) expanded view of Group 4.

**Table 1 ijms-23-03962-t001:** Gene connectivity based on protein–protein interactions computed by STRING. The orange-filled cells represent intragroup interactions, shown in Figure 1, Figure 2 and Figure 3 as edges.

Group	1	2	3	4
1	10	-	-	-
2	81	415	-	-
3	10	60	32	-
4	80	386	92	396

## Data Availability

Publicly available datasets were analyzed in this study. This data can be found here: (https://github.com/akaashvenkat/RP-Gene-Mapping).

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
