# Peer review of "Applying Protein–Protein Interactions and Complex Networks to Identify Novel Genes in Retinitis Pigmentosa Pathogenesis"

_ijms, 2022, doi:10.3390/ijms23073962_

Round 1

Reviewer 1 Report

Yoon et al. produced a very interesting article describing the “Applying Protein-Protein Interactions and Complex Networks to Identify Novel Genes in Retinitis Pigmentosa Pathogenesis”. I consider the manuscript very fascinating but, at the same time, I suggest several revisions needed to improve the reliability and the completeness of the paper:

  • The manuscript is very simple in its structure and analysis, even if effective. I suggest, at least, to seriously improve the bibliography with recent papers debating of pathway associated to retinitis pigmentosa, as results of transcriptome analyses and deep literature study. Among them, I suggest to add the recent PMID: 34440511 and PMID: 34058230.
  • Section 3.2. The authors should describe other fundamental genes involved into retinitis pigmentosa onset and progression, as they also came out from their bioinformatics analyses, such as PRPH2. Regarding this, I suggest to add the PMID: 33801777 as supporting reference.
  • Finally, manuscript requires English revisions and typos correction.

Author Response

The manuscript is very simple in its structure and analysis, even if effective. I suggest, at least, to seriously improve the bibliography with recent papers debating of pathway associated to retinitis pigmentosa, as results of transcriptome analyses and deep literature study. Among them, I suggest to add the recent PMID: 34440511 and PMID: 34058230.

 Our response: The two papers are cited, and the pathways associated with retinitis pigmentosa (RP) are discussed in the introduction.

Section 3.2. The authors should describe other fundamental genes involved into retinitis pigmentosa onset and progression, as they also came out from their bioinformatics analyses, such as PRPH2. Regarding this, I suggest to add the PMID: 33801777 as supporting reference.

Our response: The paper is cited. However, Section 3.2 is intended to focus on the novel intermediate genes rather than the known RP genes. Therefore, we would like to keep the section 3.2 same to avoid deviating from the focus of the paper.

Finally, the manuscript requires English revisions and typos correction.

Our response: Typos and a few grammatical errors are fixed.

Reviewer 2 Report

Manuscript submitted by Yoon et. al "Applying Protein-Protein Interactions and Complex Networks to Identify Novel Genes in Retinitis Pigmentosa Pathogenesis" in International Journal of Molecular Sciences (ijms-1636340). The author claims that this paper will provide better understanding of PPI and methodology employed to detect PPI. I have been carefully read the manuscript and these points should be addressed in the manuscript, as explained below:

  1. Figure 1: visibility is not good. Author must improve the image quality.
  2. Figure 3: Not well explained in the result section. Author should improve.
  3. Page 2, line 48-54: the cited reference number #10 repeated three times. Author should cite the suggested ref: Curr Protein Pept Sci. 2018, 19, 946-955.
  4. Page 2, line 78-79: “Under this notion, the genes that lack PPI to connect with any other causal genes found on the initial list may have an intermediate gene that facilitates the interaction”. To support this sentence with literature the author should cite the article: Curr Protein Pept Sci. 2018, 19, 946-955.
  5. First sentence of "conclusion" needs revision. Make it like this: In depth knowledge of the structural biology of PPIs as well as their intricate network is the key to understanding of interactions which are the very foundation of cellular processes.

Author Response

  1. Figure 1: visibility is not good. Author must improve the image quality.

 Our response: Figure 1B has been regenerated.

  1. Figure 3: Not well explained in the result section. Author should improve.

Our response: The text has been modified accordingly.

  1. Page 2, line 48-54: the cited reference number #10 repeated three times. Author should cite the suggested ref: Curr Protein Pept Sci. 2018, 19, 946-955.

 Our response:  The issue has been fixed, and the paper is cited.

  1. Page 2, line 78-79: “Under this notion, the genes that lack PPI to connect with any other causal genes found on the initial list may have an intermediate gene that facilitates the interaction”. To support this sentence with literature the author should cite the article: Curr Protein Pept Sci. 2018, 19, 946-955.

 Our response: The paper is cited.

  1. First sentence of "conclusion" needs revision. Make it like this: In depth knowledge of the structural biology of PPIs as well as their intricate network is the key to understanding of interactions which are the very foundation of cellular processes.

Our response: The sentence has been modified accordingly.